# Comparison of CoW/SiO_2_ and CoB/SiO_2_ Interconnects from the Perspective of Electrical and Reliability Characteristics

**DOI:** 10.3390/ma16041452

**Published:** 2023-02-09

**Authors:** Yi-Lung Cheng, Kai-Hsieh Wang, Chih-Yen Lee, Giin-Shan Chen, Jau-Shiung Fang

**Affiliations:** 1Department of Electrical Engineering, National Chi Nan University, Puli, Nantou 54561, Taiwan; 2Department of Materials Science and Engineering, Feng Chia University, Taichung 40724, Taiwan; 3Department of Materials Science and Engineering, National Formosa University, Huwei, Yunlin 63201, Taiwan

**Keywords:** cobalt (Co), cobalt-tungsten (CoW), cobalt-boron (CoB), electrical resistivity, barrier, reliability, time-dependent-dielectric-breakdown

## Abstract

As the feature size of integrated circuits has been scaled down to 10 nm, the rapid increase in the electrical resistance of copper (Cu) metallization has become a critical issue. To alleviate the resistance increases of Cu lines, co-sputtered CoW and CoB alloying metals were investigated as conductors and barriers in this study. Annealing CoM (M = W or B)/SiO_2_/*p*-Si structures reduced the resistivity of CoM alloys, removed sputtering-deposition-induced damage, and promoted adhesion. Additionally, both annealed CoW/SiO_2_ or CoB/SiO_2_ structures displayed a negligible V_fb_ shift from capacitance-voltage measurements under electrical stress, revealing an effective barrier capacity, which is attributed to the formation of MO_x_ layers at the CoM/SiO_2_ interface. Based on the thermodynamics, the B_2_O_3_ layer tends to form more easily than the WO_x_ layer. Hence, the annealed CoB/SiO_2_/*p*-Si MIS capacitor had a higher capacitance and a larger breakdown strength did than the annealed CoW/SiO_2_/*p*-Si MIS capacitor.

## 1. Introduction

Copper (Cu) has replaced Al as a conductive material in the back-end-of-line (BEOL) interconnect of integrated circuits owing to its lower resistivity (1.67 verses 2.67 μΩ·cm) and better electromigration resistance [1,2,3]. However, as the technology node advances to 10 nm and beyond, the resistivity of Cu lines dramatically increases due to the size effect, thereby leading to an increase in resistance-capacitance delay [4]. The size effect is related to the relatively large electron mean free path (~39 nm) of Cu, inducing severe electron scattering at the sidewalls and grain boundaries, thus increasing the line resistivity. As a result, replacing Cu with an alternative metal material that has a lower product of bulk resistivity and electron mean free path (ρ_0_ × λ) than Cu is a feasible solution. Based on this rule, Co, Ni, and Ru, with ρ_0_ × λ values comparable or even lower than that of Cu, are being considered as the potential conducting materials to replace Cu for 10 nm technology nodes and beyond [5,6].

Moreover, liner and barrier layers are required for Cu metallization interconnects in order to prevent Cu migration and promote adhesion. Traditionally, a Ta/TaN bilayer is used as a liner/barrier, which is typically deposited by physical vapor deposition. As the interconnect dimension is continuously scaled down, the poor step coverage of the physical vapor deposited on the Ta/TaN bilayer, along with its high electrical resistivity, exceeding tens of μΩ·cm, not only limits the space for Cu to fill, but it also increases the overall resistivity of the Cu interconnects [7,8]. As a result, various alternative materials, such as TaSi_x_, CoTa, CoTi_x_, and CoW, have been proposed to replace Ta/TaN as the liner and barrier layers [9,10,11,12]. These potential candidates are typically alloying metals with a single layer, which could improve barrier properties by forming an amorphous structure or by stuffing the grain boundaries.

Here, two types of Co-alloying metals, namely CoW and CoB, are proposed as conductors and barrier layers in this study. CoW alloys have been demonstrated to act as both liner and barrier layers for replacing Ta/TaN [13,14,15]. The CoW alloy exhibits an amorphous-like phase, which can eliminate diffusion path through the grain boundaries. On the other hand, the CoB alloy has rarely been studied for interconnecting materials. Faria et al. indicated that CoB can form α-Co solid solution and Co_3_B when the B concentration is ≤25 atomic% [16]. For use as interconnecting materials, the formation of a Co_3_B compound should be avoided [17]. In order to further understand CoW and CoB alloys, this study investigated the electrical and reliability characteristics of CoW/SiO_2_ and CoB/SiO_2_ interconnects under thermal and electrical stress. Adhesion and barrier properties were also studied.

## 2. Experiments

In the experiment, 100-nanometer-thick, thermally grown SiO_2_ was coated on boron-doped *p*-type Si substrates. Then, Co and CoB alloys were deposited via the co-sputtering deposition method in a magnetron sputtering system using pure Co, W, and B targets. Co was deposited by using a DC power of 100 W, and W and B were deposited by various RF powers ranging from 10 to 30 W. Before deposition, all targets were pre-sputtered for 10 min to remove surface contaminations. During the deposition of the CoM (M = W or B) film, the base pressure in the deposition chamber was 5 × 10^−6^ torr, and the working pressure was 4 × 10^−3^ torr, with a fixed Ar flow-rate of 20 sccm. The distance between the targets and the substrate was 10 cm, and the substrate was without intentional heating during sputtering. The thickness of all films was controlled at 100.0 ± 3.0 nm, as determined by an alpha-step 200 profilometer. The concentration of the alloy metal M was determined using inductively coupled plasma mass spectrometry (ICP-MS, Agilent 7500ce, Santa Clara, CA, USA). After deposition, both CoM/SiO_2_/*p*-Si samples were annealed at 425 °C for 2 h using a quartz tube furnace with an Ar + 5% H_2_ gas atmosphere.

The resistivity of the alloyed Co films was measured via the four-probe method. Transmission electron microscopy (TEM; 300 kV, Hitachi HD-2300A, Tokyo, Japan) was used to investigate the microstructure and cross-sectional profile before and after thermal annealing. The adhesion was evaluated by using tape and stud-pull tests. A commercial tape having an adhesive strength of 5.0 N/cm was adhered to the film surface, followed by a peeling-off test. Then, the number of delaminated samples was calculated.

The electrical and reliability characteristics of CoW/SiO_2_ and CoB/SiO_2_ were measured using metal–insulator–silicon (MIS) capacitors. The MIS capacitors were fabricated through a metal mask to define the top metal electrodes. The top metal electrodes were square-shaped with an area of 9.0 × 10^−4^ cm^2^. Capacitance-voltage (*C-V*) plots were measured using a precision impedance meter (Agilent 4284A), and current-voltage (*I-V*) and time-dependent-dielectric-breakdown (TDDB) plots were measured by using an electrometer (Agilent 4156C). A nitrogen gas purge was carried out during the measurements to avoid moisture absorption and metal gate oxidation. All measurements were performed at room temperature.

## 3. Results and Discussion

Table 1 shows the results of the tape and stud-pull tests of the CoW and CoB films on SiO_2_ before and after annealing. The Cu/SiO_2_ sample was also tested as a reference. A total of 81 square dots with 30 × 30 μm^2^ area were adhered using a commercial tape having an adhesive strength of 5 N/cm. Then, peeling-off tests were conducted, and the number of delaminated dots was calculated. As shown in Table 1, the Cu/SiO_2_ samples showed severe delamination, with a delamination rate of 86.2%. After annealing, all square dots were peeled off. For the as-deposited CoW/SiO_2_ and CoB/SiO_2_ films, all tested dots remained intact, indicating good adhesion for both alloyed films with SiO_2_. After annealing, the CoB/SiO_2_ samples still demonstrated a zero-delamination rate, while the delamination rate of the CoW/SiO_2_ samples increased to 5.0%. As the chemical reaction between the constituent elements occurred, adhesion was promoted. Hence, a better adhesion for the CoB/SiO_2_ samples is attributed to the chemical reaction between CoB and SiO_2_. Since the affinity of B to O is larger than that of W to O, B has a higher possibility of reacting with oxygen in the SiO_2_ [18]. The interfacial reaction is able to produce a bonding layer between CoB and SiO_2_, thus enhancing adhesion.

To understand the effect of the atomic ratios of M (W and B) on the level of resistivity, the resistivity values of the two CoM films with various M atomic ratios were measured. The result is shown in Figure 1a. For both the CoW and CoB films, the resistivity increased with the increasing concentration of the added alloying metal. The increase ratios were estimated to be 7.55 and 47.22 cm/atomic% for the CoW and CoB films, respectively, indicating that B has a larger impact than W in terms of increasing the resistivity of the Co films.

In the following experiments, the W and B atomic ratios of the CoM films were chosen so as to be similar, i.e., 4.84% and 4.71%, respectively, corresponding to the resistivity of 203.27 μΩ·cm for the CoW film and of 361.20 μΩ·cm for the CoB film. Figure 1b shows the change in the electrical resistivity of the CoW and CoB films before and after annealing at 425 °C. After annealing, the resistivity of both CoW and CoB films decreased, obviously owing to crystallization, defect annihilation, and grain growth [19]. The resistivity decreased to 26.95 and to 23.86 μΩ·cm for the CoW and CoB films, respectively, which is much higher than the reported 5.7 μΩ·cm of Co film [20]. This result indicates that the CoB films exhibited a larger decrease in the resistivity upon annealing. Via annealing, the resistivity of the CoB films was made similar to that of the CoW films even though the resistivity of the as-deposited CoB films is higher.

Figure 2a–c shows the cross-sectional TEM images of the CoW/SiO_2_ and CoB/SiO_2_ samples before and after 425 °C annealing for 1 h. For the as-deposited samples, both the CoW and CoB films were amorphous. Additionally, both samples had a visible interface between the CoM and SiO_2_ films, without the formation of an additional layer. After annealing at 425 °C, both the CoW and CoB films crystallized and were transformed into fine equiaxed grains. For the annealed CoW/SiO_2_ sample, the interface between the CoW and SiO_2_ films remained unchanged. On the other hand, after the same annealing, an ultrathin layer with an estimated thickness of 5.24 ± 0.28 nm was observed at the interface between the CoB and SiO_2_ films. This layer likely formed via an annealing-induced chemical interaction between B and the oxygen in the SiO_2_ film.

The measurements of the negative heat of WO_2_ and WO_3_ formation at 25 °C are −295 kJ/mole and −281 kJ/mole, respectively, per oxygen atom, and that of B_2_O_3_ is −421 kJ/mole per oxygen atom. The W- and B-based oxides each has a lower formation heat than does SiO_2_ (−455 kJ/mole per oxygen atom) [18], representing that SiO_2_ is more thermally stable. Hence, the W and B in the as-deposited samples could not reduce the surrounding SiO_2_ and form M-oxide thermodynamically. Upon annealing, the thermal energy drives the reaction between B and SiO_2_ to occur, thereby forming a layer of B_2_O_3_ at the CoB/SiO_2_ interface, as evidenced in Figure 2c. Since the formation heats of WO_2_ and WO_3_ are far lower than that of SiO_2_, the reaction between W and SiO_2_ cannot occur, even with annealing at 425 °C. As a result, no interfacial layer can be observed in the annealed CoW/SiO_2_ sample.

Figure 3 compares the *C*-*V* curves of the CoW-gate and CoB-gate MIS capacitors before and after annealing. For both the CoW-gate and CoB-gate MIS capacitors, a larger accumulation capacitance (*C*_acc_) and negative voltage shift in the *C*-*V* curves were observed in the as-deposited samples. Additionally, the CoB-gate MIS capacitor had a higher *C*_acc_ and a larger voltage shift than did the CoW-gate samples. The result implies that the properties of MIS capacitors may be modified by the sputtering deposition of metal gates. Following thermal annealing, the *C*_acc_ values for both the CoW-gate and CoB-gate MIS capacitors were reduced. Moreover, the *C*-*V* curves turned towards the right direction, with flat-band voltages (*V*_fb_) of about −0.6 V for both samples. In this stage, the CoB-gate MIS capacitor still had a higher *C*_acc_ than that of the CoW-gate sample.

Based on the measured *C*_acc_, the dielectric constant (*k*) of the SiO_2_ film in the MIS capacitor can be determined by using *k* = *C_acc_d*/ε_0_*A*. Here, ε_0_ is vacuum permittivity, *d* is film thickness, and *A* is the gate area. Figure 4 shows the *k* values of the SiO_2_ film in the CoW-gate and CoB-gate MIS capacitors before and after 425 °C annealing. The theoretical *k* value of the thermally grown SiO_2_ film is 3.9~4.1. For the as-deposited CoW-gate and CoB-gate MIS capacitors, the *k* values of the SiO_2_ films were determined to be 5.58 ± 0.32 and 6.07 ± 0.54, respectively. An increased *k* value demonstrates that the SiO_2_ film was modified during the sputtering deposition of the metal gates. During the sputtering deposition of the CoM alloys, Ar gas was ionized in the plasma environment. Vacuum ultraviolet light was also emitted. These plasma-generated species could damage the SiO_2_ film’s surface and produce trapped charges within the film [21]. This speculated mechanism can be confirmed by the large shifts of the *C*-*V* curves in the as-deposited samples. The V_fb_ values of the as-deposited CoW-gate and CoB-gate MIS capacitors shifted to −6.26 V and −10.75 V, respectively, representing that positive charges were produced and trapped in the SiO_2_ film. The sputtering deposition of the CoB alloys produced more positive charges than that of the CoW alloys.

Annealing reduced the *k* values of the SiO_2_ films in both MIS capacitors, indicating recovery of the dielectric properties of the SiO_2_ film via the annealing. The *k* value of the SiO_2_ film in the annealed CoW-gate MIS capacitors was reduced to 4.12 ± 0.17, which is similar to the theoretical value of thermally grown SiO_2_ films. This result suggests that most of sputtering-deposition-induced damage was repaired by thermal annealing, similar to the results of our previous study [22]. For the CoB-gate MIS capacitors, the *k* value of the SiO_2_ film was reduced to 4.72 ± 0.22 by thermal annealing, but it was still higher than the theoretical value. The higher *k* value of the SiO_2_ film in the annealed CoB-gate MIS capacitors is likely attributable to the formation of an interfacial B_2_O_3_ layer, which has a higher dielectric constant than SiO_2_ [23].

Moreover, the *V*_fb_ values of both the CoW-gate and CoB-gate MIS capacitors became ~−0.6 V after annealing. The *V*_fb_ value of an MIS capacitor is determined by the work function difference between the metal gate and the Si, provided that the used dielectric film has no charges. The theoretical work functions of *p*-type Si and Co are 5.25 eV and 5.00 eV, respectively [18]. Hence, the V_fb_ values of CoW-gate and CoB-gate MIS capacitors are estimated to both be approximately −0.25 eV. The dielectric film in the MIS capacitors is thermally grown SiO_2_, which reportedly has a typical fixed-charge density on the order of 10^10^ to 10^11^ cm^−2^. These fixed charges have been reported to be positive charges due to the form of oxygen vacancies, giving rise to a *V*_fb_ shift of approximately −0.01 to −0.1 V [24,25]. Herein, annealing caused the *V*_fb_ values of both the CoW-gate and CoB-gate MIS capacitors to reduce to −0.6 V, very close to the theoretical value. This result indicates that the sputtering-induced charges in the SiO_2_ film were passivated by annealing. Furthermore, annealing did not cause further negative *V*_fb_ shift, representing no diffusion of metal ions into the SiO_2_. This result reveals that CoW and CoB alloys can act as a diffusion barrier in advanced integrated circuits.

Figure 5 presents the plots of the leakage current-electric field (*I-E*) curves of the annealed CoW-gate and CoB-gate MIS capacitors. Ten samples were measured for each condition. From the *I-E* plots, the breakdown field can be determined as the leakage current suddenly increases by at least three orders of magnitude to more than 10^−2^ A. Before breakdown, the measured leakage current of the annealed CoW-gate MIS capacitor was ~10^−10^A, and it and remained unchanged with the applied field, which is similar to the behavior of the annealed Co-gate MIS capacitor [26]. For the annealed CoB-gate MIS capacitor, its current increased with the applied field and was larger than that of annealed CoW-gate MIS capacitor. This transition is likely to be the formation of the interfacial B_2_O_3_ layer, which increases the leakage current.

Additionally, the breakdown field of the annealed CoB-gate MIS capacitor was 10.37 ± 0.55 MV/cm, which was higher than that of the annealed CoW-gate sample (9.72 ± 0.35 MV/cm). An increased breakdown field for the annealed CoB-gate MIS capacitor is attributed to the formation of B_2_O_3_ interfacial layer. This self-forming layer, due to thermal annealing, provides an additional resistance to failure.

Electrical stress was applied to the annealed CoW-gate and CoB-gate MIS capacitors. After the application of electrical stress, *C-V* measurements were taken. Figure 6a,b show the plots of the normalized *C-V* curves for the annealed CoW-gate and CoB-gate MIS capacitors, respectively, after they were subjected to electrical stress at ±9.0 MV/cm for 10^3^ s. Under both positive-polarity and negative-polarity electrical stress, the *C-V* cures shifted to a negative voltage direction for both the annealed CoW-gate and the CoB-gate samples. A larger shift was observed under negative-polarity electrical stress. Figure 7 summarizes the *V*_fb_ shifts as a function of the applied field for the annealed CoW-gate and CoB-gate MIS capacitors after they were subjected to electrical stress for 10^3^ s. Whether under electrical stress with various fields in the positive- polarity or the negative-polarity, both annealed CoW-gate and CoB-gate MIS capacitors exhibited negative *V*_fb_ shifts, indicating that positive charges were introduced into the SiO_2_ film. The magnitude of the *V*_fb_ shift gradually increased with the increasing stress field, and it was larger with the negative-polarity electrical stress than with the positive-polarity electrical stress. Additionally, the annealed CoW-gate MIS capacitor had a larger *V*_fb_ shift than did the annealed CoB-gate sample. The *V*_fb_ shift for the annealed CoW-gate MIS capacitor became pronounced as the stress field increased to −9.0 MV/cm. Under positive-polarity electrical stress, metal ions would drift into a dielectric film, thus causing a negative *V*_fb_ shift [18,27]. As a result, the negative *V*_fb_ shift for the annealed CoW-gate and CoB-gate MIS capacitors under positive-polarity electrical stress was likely caused by the drifting metal ions. In the case of the annealed Co-gate MIS capacitor, which underwent the identical electrical stress at 9.0 MV/cm, the *V*_fb_ shift was −4.25 V, which was higher than those of the annealed CoW-gate and CoB-gate samples. This indicates that the doping element (W or B) in the Co can prevent the drift of Co ions. Comparing the annealed CoW-gate and CoB-gate samples reveals that the latter had a smaller negative V_fb_ shift, indicating better resistance to prevent the migration of metal ions. This prevention is attributed to the formation of an interfacial B_2_O_3_ layer in the annealed CoB-gate sample.

Time-dependence-dielectric-breakdown (TDDB) tests were carried out to evaluate the long-term reliability of the annealed CoW-gate and CoB-gate MIS capacitors. In a TDDB test, negative-polarity electrical stress was continuously applied to the metal gate of an MIS capacitor, and the response leakage current was continuously monitored with the stressing time. Figure 8 shows the plots of the leakage current versus the stressing time (*I-t*) for the annealed CoW-gate and CoB-gate MIS capacitors during the TDDB tests. The MIS capacitors with different metal gates exhibited very different characteristics in the *I-t* curves. The curves from the annealed CoW-gate sample showed that the current increased with the stressing time and then suddenly jumped (i.e., broke down). This gradually increased current is attributed to charge trapping in the SiO_2_ layer. As the number of trapped charges reaches the critical value, breakdown occurs due to the formation of a conduction path. In contrast, the curves from the annealed CoB-gate sample had almost flat *I-t* curves, suggesting that no charges were trapped in the SiO_2_ layer. The differences in behavior of the *I-t* curves for the annealed CoW-gate and CoB-gate MIS capacitors are likely to be caused by the CoM/SiO_2_ interface. The interfacial B_2_O_3_ layer is formed in the CoB/SiO_2_ structure after annealing. This layer helps to prevent charge-trapping in the SiO_2_ layer.

Figure 9 shows the plots of the breakdown times (*t*_bd_) of the annealed CoW-gate and CoB-gate MIS capacitors as a function of the stressing electric-field. The breakdown times of both the annealed CoW-gate and CoB-gate MIS capacitors decreased with increases in the applied field, indicating that the field plays an important role in controlling dielectric breakdown. Comparing the breakdown times of the annealed CoW-gate and CoB-gate MIS capacitors reveals that the latter sample had longer breakdown times, similar to the results of the breakdown field. The formation of a thick interfacial B_2_O_3_ layer and fewer electric-stress-induced charges in the annealed CoB-gate MIS capacitor are believed to be responsible for the enhanced TDDB reliability.

## 4. Conclusions

The electrical and reliability characteristics of CoW and CoB films on SiO_2_/*p*-Si substrates were investigated in this study. Both CoW and CoB films provided enhanced adhesion with SiO_2_, and thus a better diffusion barrier against Co diffusion. After annealing, the reaction between M (W or B) and SiO_2_ occurred, and a thin MO_x_ layer was formed. Based on the thermodynamics, the B_2_O_3_ layer tends to form more easily than does the WO_x_ layer, which is consistent with the TEM results. As a result, the annealed CoB/SiO_2_/*p*-Si MIS samples displayed larger breakdown fields and longer TDDB breakdown times than the annealed CoW/SiO_2_/*p*-Si MIS samples, indicating that, from the viewpoint of electrical and reliability characteristics, the CoB alloy is a promising candidate for diffusion-barrier application in advanced interconnects.

## Figures and Tables

**Figure 1 materials-16-01452-f001:**
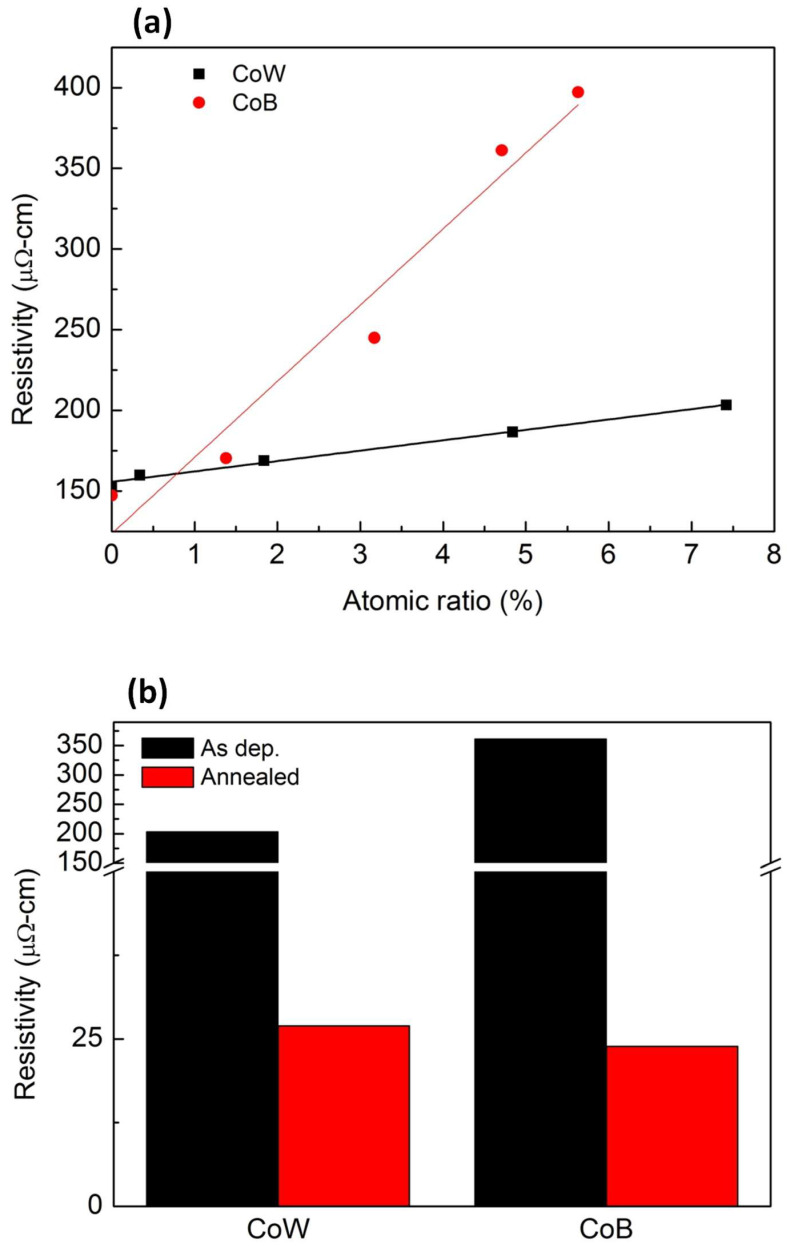
(**a**) Resistivity of CoW and CoB films as a function of alloying-element concentration; (**b**) resistivity changes in CoW and CoB films after annealing at 425 °C.

**Figure 2 materials-16-01452-f002:**
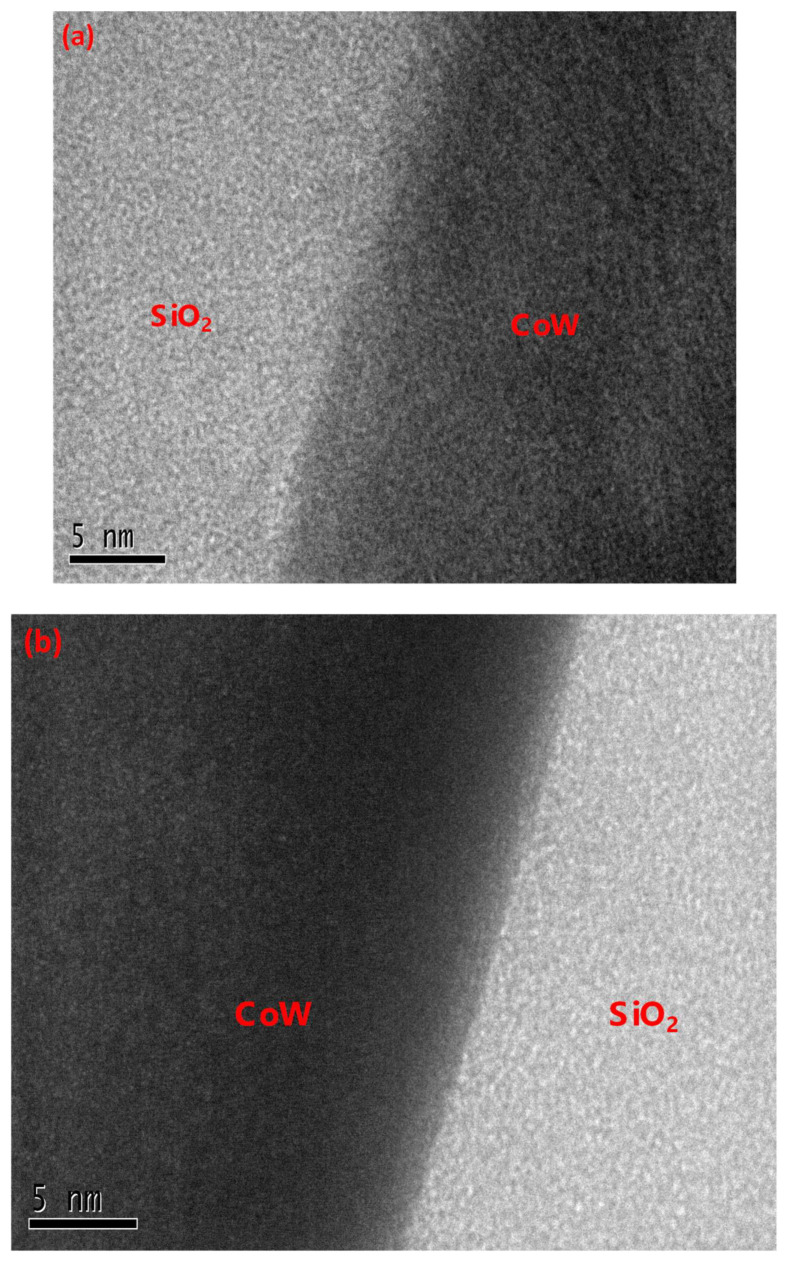
Cross-sectional TEM images: (**a**) as-deposited CuW/SiO_2_/Si; (**b**) annealed CuW/SiO_2_/Si; (**c**) annealed CuB/SiO_2_/Si.

**Figure 3 materials-16-01452-f003:**
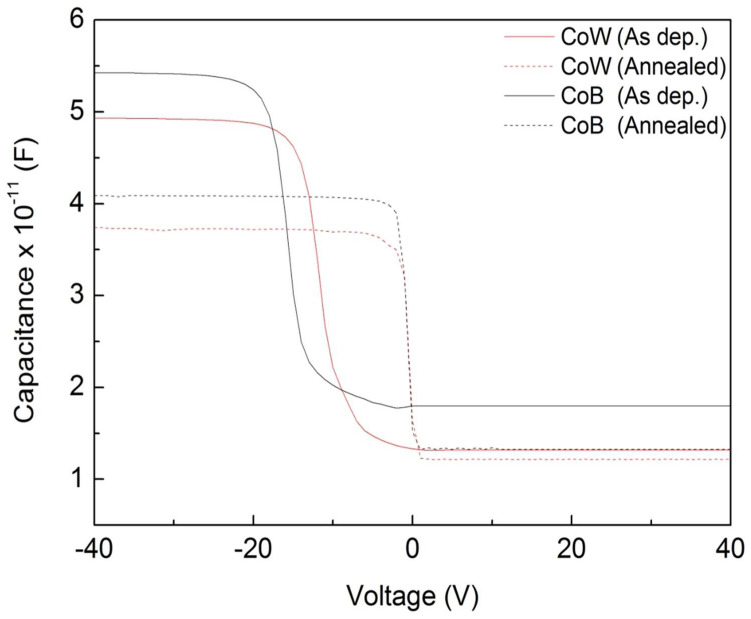
*C-V* curves of CoW-gate and CoB-gate MIS capacitors before and after annealing.

**Figure 4 materials-16-01452-f004:**
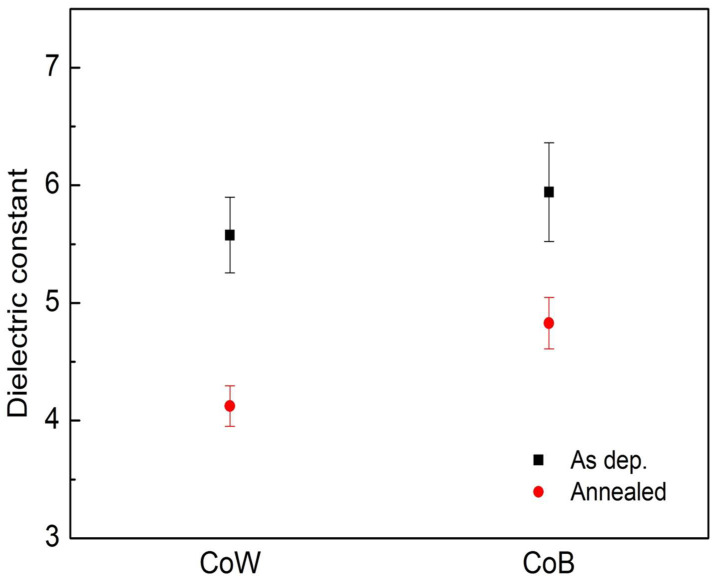
Dielectric constants of SiO_2_ films in CoW-gate and CoB-gate MIS capacitors before and after annealing.

**Figure 5 materials-16-01452-f005:**
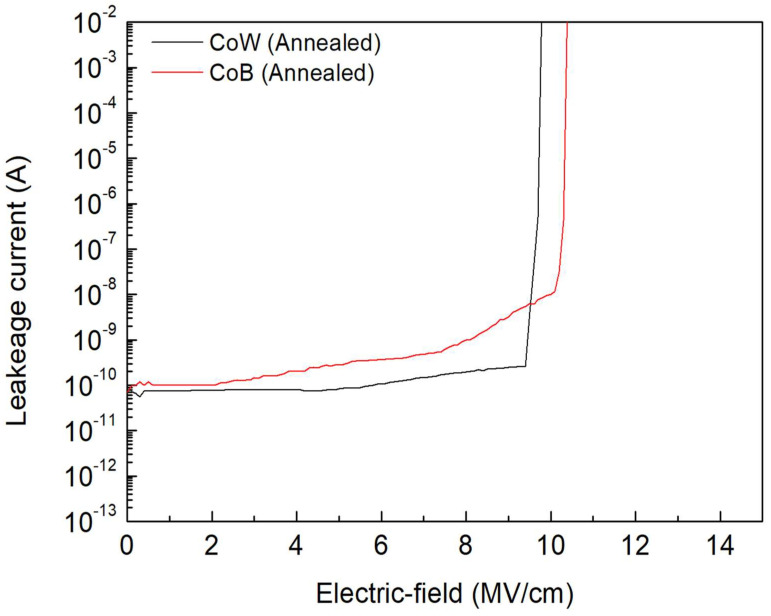
*I-E* plots of annealed CoW-gate and CoB-gate MIS capacitors.

**Figure 6 materials-16-01452-f006:**
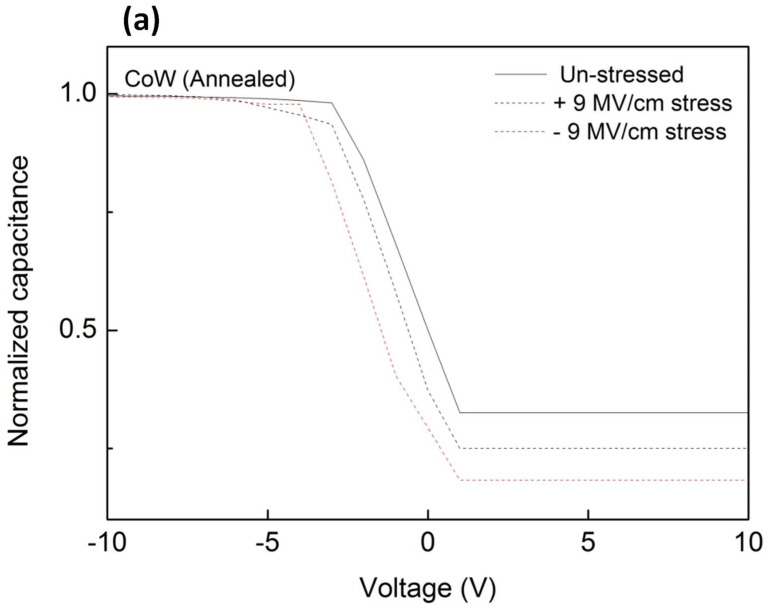
*C-V* curves of annealed MIS capacitors after electrical stress at ±9.0 MV/cm for 10^3^ s; (**a**) CoW-gate; (**b**) CoB-gate.

**Figure 7 materials-16-01452-f007:**
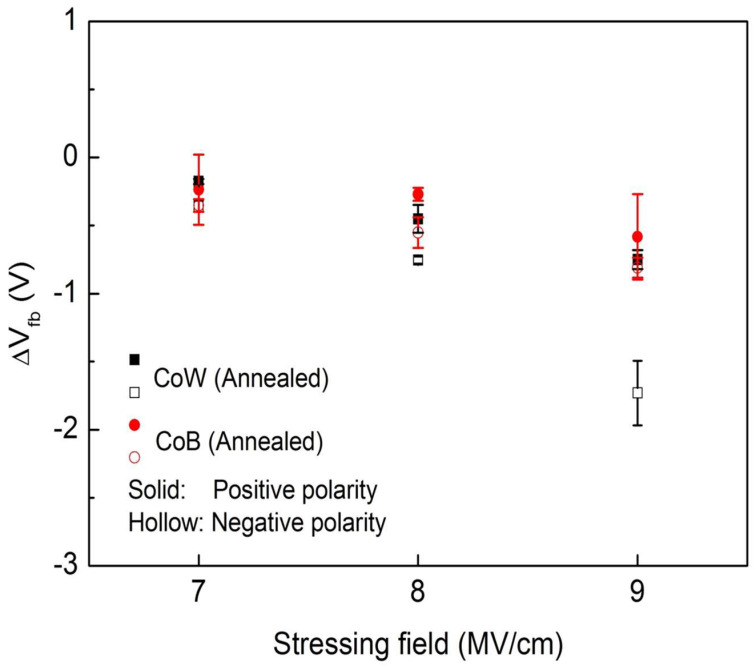
V_fb_ shifts of annealed CoW-gate and CoB-gate MIS capacitors after electrical stress as a function of stress field.

**Figure 8 materials-16-01452-f008:**
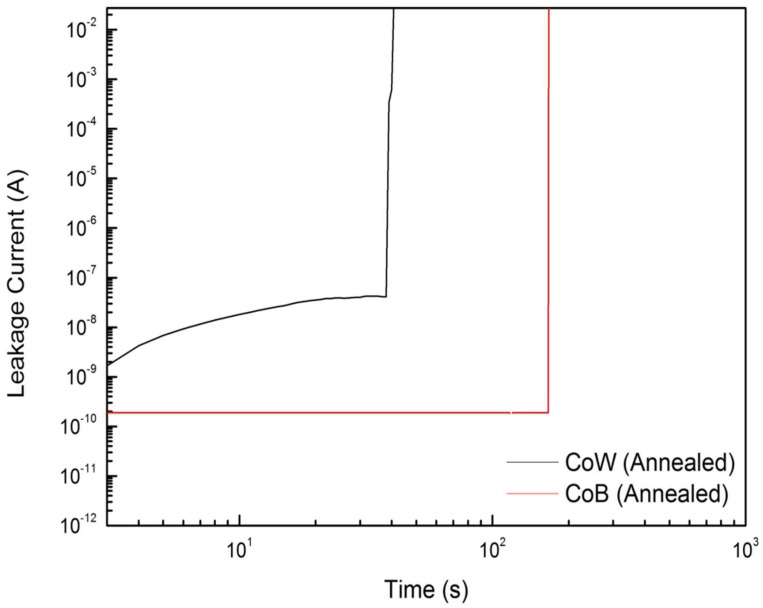
Leakage current versus stressing time for annealed CoW-gate and CoB-gate MIS capacitors during TDDB tests.

**Figure 9 materials-16-01452-f009:**
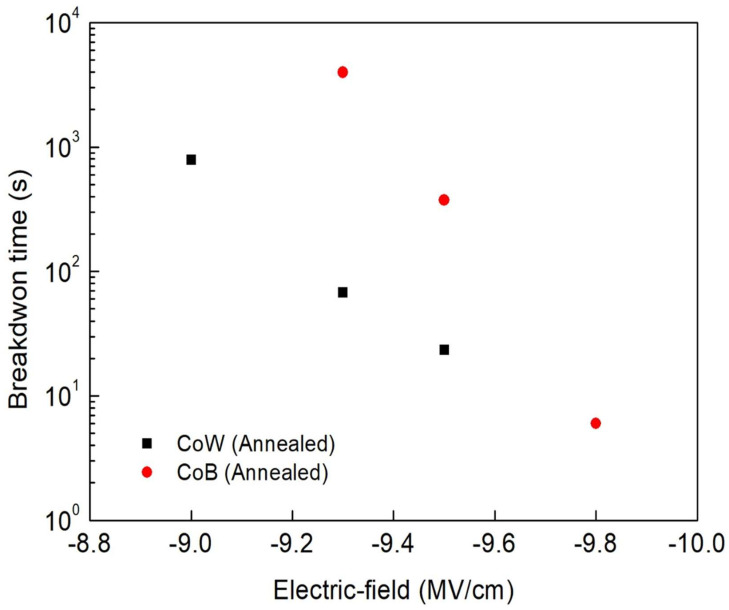
Comparison of breakdown times for annealed CoW-gate and CoB-gate MIS capacitors as a function of stressing field.

**Table 1 materials-16-01452-t001:** Tape test results for Cu/SiO_2_, CoW/SiO_2_, and CoB/SiO_2_ before and after annealing at 425 °C.

	As Dep.	Annealed
Cu/SiO_2_	86.2%	100%
CoW/SiO_2_	0%	5.0%
CoB/SiO_2_	0%	0%

## Data Availability

Not applicable.

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
