# Peer review of "Comparison of CoW/SiO2 and CoB/SiO2 Interconnects from the Perspective of Electrical and Reliability Characteristics"

_materials, 2023, doi:10.3390/ma16041452_

Round 1

Reviewer 1 Report

The authors should address the following:

1. The abstract is poorly presented and doesn't reflect the contribution of the manuscript

2. Figures, tables, typos,... all must be corrected and presented professionally and consistently 

3. Regarding the results reported in figure 5, it is not clear how many measurements were performed in order to obtain that figure

4. do you really need a separate figure (6) in order to report two values?!! The article must be improved. 

5. the variation of capacitance vs voltage doesn't look convincing. The measured values must be tabulated so that an expert can decide on the correctness of the measurements

Author Response

  1. The abstract is poorly presented and doesn't reflect the contribution of the manuscript

[Reply] The abstract had been revised and the contribution of the manuscript had been mentioned.

  1. Figures, tables, typos,... all must be corrected and presented professionally and consistently 

[Reply] Grammatical and writing style errors in the original version had been corrected again by our colleague who is a native English speaker. In addition, the figures had been re-plotted and revised in order to improve the quality and make it clearer.

  1. Regarding the results reported in figure 5, it is not clear how many measurements were performed in order to obtain that figure

[Reply] The measurement points had been added in the revised manuscript. “Ten samples were measured for each condition.

  1. do you really need a separate figure (6) in order to report two values?!! The article must be improved. 

[Reply] Figure (6) had been deleted and the result of Fig.6 had been reported in the text in the revised manuscript.

  1. the variation of capacitance vs voltage doesn't look convincing. The measured values must be tabulated so that an expert can decide on the correctness of the measurements

[Reply] Fig.3 and Fig.4 had displayed the variation of capacitance.

Reviewer 2 Report

Authors in this research article have presented and investigated Comparison of CoW/SiO2 and CoB/SiO2 interconnects from perspective of electrical and reliability characteristics. This article is suitable for publication in this journal after edit language.

Author Response

Authors in this research article have presented and investigated Comparison of CoW/SiO2 and CoB/SiO2 interconnects from perspective of electrical and reliability characteristics. This article is suitable for publication in this journal after edit language.

[Reply] Thanks for reviewer's positive comment!

Reviewer 3 Report

The article titled Comparison of CoW/SiO2 and CoB/SiO2 interconnects from perspective of electrical and reliability characteristics describes very interesting topic, ut before publication English should be checked. Moreover, the literature positions need to be refilled with newer items from the last 10 years. 

Author Response

 The article titled Comparison of CoW/SiO2 and CoB/SiO2 interconnects from perspective of electrical and reliability characteristics describes very interesting topic, ut before publication English should be checked. Moreover, the literature positions need to be refilled with newer items from the last 10 years. 

[Reply] Grammatical and writing style errors in the original version had been corrected again by our colleague who is a native English speaker. In addition, the literature had been revised with newer items from the last 10 years.

Reviewer 4 Report

Dear Authors,

Thank you for submitting your manuscript to Materials journal.

Please find attached your pdf manuscript with comments for minor revision.

Good luck!

Author Response

Thank you for submitting your manuscript to Materials journal.

Please find attached your pdf manuscript with comments for minor revision.

Good luck!

[Reply] The reviewer’s comments had been revised point by point.

Reviewer 5 Report

  • Review:
    This research paper is focused on comparison between CoW/SiO2 and CoB/SiO2 as interlayers of choice for an advanced integrated circuit technology. After presenting the state of the art in the introduction, authors described the preparation procedure and the selected setup for structural and electrical characterization. In results and discussion, they gave enough arguments to conclude that the CoB/SiO2 combination presents a better barrier. Such conclusion should have been more clearly stated in the conclusion.  
  • The importance and soundness of the proposed hypotheses;
    OK
  • The suitability and feasibility of the experimental and analysis methodology;
    The selected analytical techniques are adequate to test the hypothesis.
  • Whether there are sufficient details given to replicate the proposed experimental procedures and analysis;
    YES

Specific comments
- page 2 last §.
a delamination rate of 84.0 %... Table 1. states 86.2%!!!!!
- page 3 1
§. To understand atomic ratios of M (W and B) on the resistivity… verb missing!!
- page 3 1
§. … indicating that B has a smaller impact than W in terms of resistivity increasing of the Co films.  Is this right, Fig 1a shows opposite?
-
ratios of the Co-alloy films were chosen to be similar, i.e., 4.84 % and 4.71 %, respectively, corresponding to the resistivity of 203.27 μΩ-cm for CoW film and 361.20 μΩ-cm for CoB film. Fig. 1 (b)….. resistivity values in Fig 1B are different!

Author Response

 This research paper is focused on comparison between CoW/SiO2 and CoB/SiO2 as interlayers of choice for an advanced integrated circuit technology. After presenting the state of the art in the introduction, authors described the preparation procedure and the selected setup for structural and electrical characterization. In results and discussion, they gave enough arguments to conclude that the CoB/SiOcombination presents a better barrier. Such conclusion should have been more clearly stated in the conclusion.  

[Reply] The abstract and conclusion had been revised in the revised manuscript.

  • The importance and soundness of the proposed hypotheses;
    OK
  • The suitability and feasibility of the experimental and analysis methodology;
    The selected analytical techniques are adequate to test the hypothesis.
  • Whether there are sufficient details given to replicate the proposed experimental procedures and analysis;
    YES

Specific comments
- page 2 last §. a delamination rate of 84.0 %... Table 1. states 86.2%!!!!!
- page 3 1 §. To understand atomic ratios of M (W and B) on the resistivity… verb missing!!
- page 3 1 §. … indicating that B has a smaller impact than W in terms of resistivity increasing of the Co films.  Is this right, Fig 1a shows opposite?
- ratios of the Co-alloy films were chosen to be similar, i.e., 4.84 % and 4.71 %, respectively, corresponding to the resistivity of 203.27 μΩ-cm for CoW film and 361.20 μΩ-cm for CoB film. Fig. 1 (b)….. resistivity values in Fig 1B are different!

[Reply] All errors had been corrected.

Round 2

Reviewer 1 Report

The comments are addressed.